# Spatial Differences in the Chemical Composition of Surface Water in the Hornsund Fjord Area: A Statistical Analysis with A Focus on Local Pollution Sources

**Krystyna Koziol** [1] , **Marek Ruman** [2,*] , **Filip Pawlak** [1] , **Stanisław Chmiel** [3] and **Żaneta Polkowska** [1]

[1] Department of Analytical Chemistry, Chemical Faculty, Gdańsk University of Technology, G. Narutowicza 11/12, 80-233 Gdańsk, Poland; krykozio@pg.edu.pl (K.K.); filpawla@student.pg.edu.pl (F.P.); zanpolko@pg.edu.pl (Ż.P.)

[2] Institute of Earth Sciences, Faculty of Natural Sciences, University of Silesia, Będzińska 60, 41-200 Sosnowiec, Poland

[3] Department of Hydrology and Climatology, Faculty of Earth Sciences and Spatial Management, Maria Curie-Skłodowska University, Krasnicka 2d, 20-718 Lublin, Poland; stanislaw.chmiel@poczta.umcs.lublin.pl

* Correspondence: marek.ruman@us.edu.pl

**Abstract:** Surface catchments in Svalbard are sensitive to external pollution, and yet what is frequently considered external contamination may originate from local sources and natural processes. In this work, we analyze the chemical composition of surface waters in the catchments surrounding the Polish Polar Station in Svalbard, Hornsund fjord area. We have pooled unpublished and already published data describing surface water composition in 2010, related to its pH, electrical conductivity (EC), metals and metalloids, total organic carbon (TOC) and selected organic compound concentrations, including persistent organic pollutants (POPs) and surfactants. These data were statistically analyzed for spatial differences, using Kruskal–Wallis ANOVA and principal component analysis (PCA), with distance from the station in the PCA approximating local human activity impact. The geological composition of the substratum was found to be a strong determinant of metal and metalloid concentrations, sufficient to explain significant differences between the studied water bodies, except for the concentration of Cr. The past and present human activity in the area may have contributed also to some of the polycyclic aromatic hydrocarbons (PAHs), although only in the case of naphthalene can such an effect be confirmed by an inverse correlation with distance from the station. Other likely factors contributing to the chemical concentrations in the local waters are marine influence, long-range pollution transport and release from past deposition in the environment.

**Keywords:** trace elements; POPs (persistent organic pollutants); surfactants; organic ions; Arctic; lake; stream; local contamination

---

## 1. Introduction

Svalbard is a remote archipelago, yet it is exposed to long-range transported contamination. While the exposure to external factors has been known for a long time (e.g., [1,2]), the local pollution there may be more important in some locations, such as settlements [3,4] which were typically established as mining towns. The mining, power production, transport, waste and sewage disposal all contribute to the local pollution [5]. However, there remain knowledge gaps regarding the exact impact of the local human activities, especially with respect to their spatial coverage, the chemical

contamination profile, and temporal trends. A non-mining settlement, such as the Hornsund research station, is an important point on the pollution map of Svalbard to gain a full picture of it.

The vicinity of the Polish Polar Station (PSP) has been subject to multiple chemical studies of the surface waters [6–16]. Two streams in the vicinity of the PSP (Fuglebekken and Ariebekken in the Revelva catchment) are regularly monitored by the station crew for inorganic ion concentrations, pH and electrical conductivity (EC) (supervised by A. Nawrot; https://hornsund.igf.edu.pl/wp-content/uploads/2017/08/Wykaz-monitoringu-geosystemu-prowadzonego-w-Polskiej-Stacji-Polarnej-Hornsund.pdf). The local influence of the station has been described in the 1980s by Krzyszowska [17,18], highlighting the impact of old fuel spills upon the local soils. Recently, she has re-surveyed the area, concluding the extent of spill traces has shrunk by 50% [19]. However, the newer information is more typically provided as part of multidisciplinary studies. For example Wojtuń et al. [20] mention elevated Co, Li, and Ni concentrations in the moss *Racomitrium lanuginosum* in the vicinity of the PSP Hornsund as compared to the locations more distant from it, and they ascribe this to operating a waste incinerator in the station. Recently, a report has been published on local contamination in Svalbard [5], but little information on Hornsund is given there. Since Hornsund fjord was listed among twelve European Marine Biodiversity Flagship Sites by the 5th Framework Programme of the European Community, its wider environmental relevance justifies assessing the impact of local pollution in the area. It is also important in order to avoid extrapolating its chemical characteristics groundlessly.

The neighborhood of the station is characterized by a varied geological substratum, surface topography and geomorphology, and vegetation. Kozak et al. [10] described the geological substratum of the Fuglebekken catchment in the context of the local sources of metals and metalloids. From a literature review, they identified potential local sources of Li, Na, K, Be, Mg, Ca, Th, U, Ti, Zr, Nb, Ta, Mn, Fe, Al, and rare earth elements, possibly also Sn, Sr and Pb [21], as well as Ba and Zn. Kosek et al. [14,15] provide similar information for the Revelva catchment, listing local sources of Li, Rb, Cs, Ba, Zr, V, Cr, Mn, Fe, Cu, Zn, B, Al and Pb.

The study area is frequently subject to field studies operating from the basis at the PSP, a likely source of small chemical modifications, on the geochemical background of the landscape-derived variety. Despite the consistent efforts to limit the contamination connected to the operation of the Arctic research stations, the PSP is not a zero-emission facility. The chemical diversity introduced by all these factors is subject to investigation here.

## 2. Materials and Methods

The dataset used in this paper consists of both already published and unpublished data. The already published parts of this dataset were described in [8,10,11,16]. Kozak and others have elaborated on the pH, EC, and the concentrations of metals and metalloids, as well as the total organic carbon (TOC), measured as non-purgeable organic carbon (NPOC), in the catchments of Fuglebekken [10] and Revelva [11]. Ntougias et al. [8] report the data on metals and metalloids, surfactants, TOC, formaldehyde and total phenols concentrations in a subset of samples, the EC and pH of those, as well as the ion chemistry of those samples and their bacterial community (for both Fuglebekken and Revelva catchments, but only pertaining to eight samples with the full set of parameters measured). Finally, Pawlak et al. [16] focused on the concentrations of polycyclic aromatic hydrocarbons (PAHs) and polychlorinated biphenyls (PCBs), with the pH, EC and TOC as background information, for a set of 30 samples from streams and lakes in the vicinity of the PSP. Although important parts of the dataset have already been explored, there remained data pools virtually untouched, such as the wider surfactant concentration data and the short-chain fatty acids concentrations.

However, the most important contribution of this study is to treat the described wide dataset in full, allowing for a direct comparison between various regional water body clusters during the same season. All samples used in this study were collected in late summer 2010 (August/September), thus we avoid seasonal differences influencing the interpretation of the analysed chemical concentrations. The data,

originally collected for various projects, were pooled into this one dataset, thus the numbers of samples analyzed for various types of chemical compounds differ markedly. In total, 75 stream/river water samples and 99 lake water samples were subject to some or all analytical procedures described below.

### 2.1. Field Site

The Hornsund fjord is the southernmost fjord of the Spitsbergen island (Svalbard archipelago), located in the Norwegian Arctic. Polish Polar Station (PSP; 77° N and 15°33′ E) is situated near its outlet, at 11 m asl, near the shore. It neighbors the surface catchment of the stream Fuglebekken in the north-east and a sea-shore plain with small tundra lakes, episodic streams and mossy wetlands (Fuglebergsletta) in the west. Further west, there is a bigger surface catchment of Revelva (Figure 1).

The geological substratum of the Fuglebekken and Revelva catchments belongs to the Svalbard Archipelago proto-basement (Hecla Hoek formation) [22,23]. The Fuglebekken catchment lies upon metamorphic rocks, formed from sediments rich in quartz and silicate with a secondary admixture of carbonates [22]. The same rock formation extends across the lower part of the Revelva catchment, while its upper part is built mainly of metamorphosed silicate deposits (shists) [24]. A small area near the springs of Revelva is built of rocks with a more varied composition: quartzite, amphibolites and migmatites [24,25]. Across the Revelva catchment, ore-bearing mineral veins may be found which include iron minerals: ankerite, pyrite, chalcopyrite, pyrrhotite, sometimes also magnetite and haematite [26], and the occurrence of ore-bearing veins extends in the whole study area and beyond.

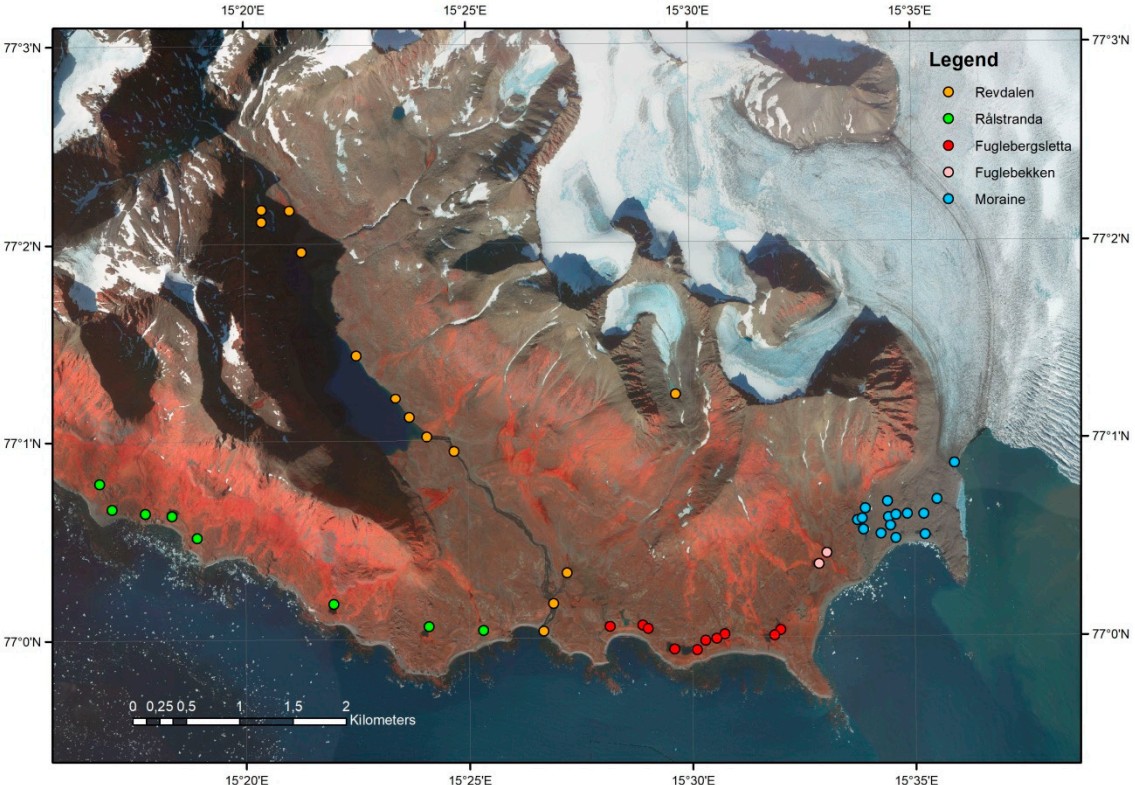

**Figure 1.** Sampling points in the Hornsund fjord area, divided into five regions (background map [27]).

Both Fuglebekken and Revelva are mixed regime watercourses, supplied by snowmelt in the early summer and permafrost thaw in the later warm season, while occasionally being predominantly fed by heavy rain [7,15]. They drain into the bays of the Hornsund fjord, Isbjørnhamna and Ariebukta, respectively. Revelva is 5.3 km long (including the lake it passes through), and is predominantly fed by left tributaries, including the proglacial Ariebekken. Both catchments, in their lower parts, form an elevated marine terrace [10,11].

During fieldwork in 2010, various new hydrological measurements have been conducted (M. Ruman, unpublished data). Water bodies of the area have been mapped, including their hydrochemical characteristics (using a multiparameter probe for the measurements of the physical and chemical characteristics of water). River discharge has been measured in the area. The Revdalen lake (Revvatnet) has been investigated in more detail, including bathymetric profiling and both water and sediment sampling at the lake bottom by scuba divers. It has been determined that Revvatnet has a groundwater-draining character, yet the residence time of water in the lake has not been estimated yet. It is likely that melting permafrost may also feed some of the other investigated lakes, yet no hydrogeological survey has been performed by the authors to confirm this hypothesis.

## 2.2. Sampling

The surface water samples analyzed in this work represent flowing and stagnant water from water bodies of various size (from puddles and small streams to a large lake and river), and they were captured into airtight high-density polyethylene (HDPE) bottles (1L or smaller). Stagnant pools and lakes were sampled in the way that prevented disturbance of bottom sediments, ideally at 50 cm below water surface and 2 m from the shore, however, in the smaller water bodies this was not possible. Flowing water was taken from the main stream at a depth of 20 cm below water level. Each bottle was triple rinsed with the sample prior to its collection. Samples were transported to the laboratory in the dark, in cold storage (approximately +4 °C). Samples were of various size (water volume) and thus it was impossible to complete all assays presented in this work for all sampling points. The limited data is a result of the samples being originally collected for a set of various studies and only afterwards the collected information was pooled for the wider spatial comparison presented here.

## 2.3. Laboratory Measurements

The measurements of basic parameters, such as EC and pH, were carried out in the freshly collected samples (using a microcomputer pH-meter and conductivity meter; inoLab®Multi 9310 (WTW, Weilheim, Germany) IDS pH), equipped with a Tetra-Con®925 (WTW, Weilheim, Germany) conductivity sensor and a SenTix®940 (WTW, Weilheim, Germany) electrode. After collection, each sample was divided into aliquots for separate determination procedures: metals and metalloids (Al, As, B, Ba, Cd, Co, Cs, Cr, Cu, Li, Mn, Mo, Ni, Pb, Rb, Sb, Se, Sr, Th, Tl, U, V and Zn), persistent organic pollutants (POPs: PAHs and PCBs), surfactants (anionic, cationic and nonionic), and other organic compounds and summary parameters (TOC, formaldehyde, sum of phenols, SCFAs = short-chain fatty acids: acetate, propionate, butyrate, oxalate, citrate, malonate, valerate and isovalerate).

The metals and metalloids were determined using ICP-MS (inductively coupled plasma mass spectrometry; Elan DRC), with argon fed to the atomizer (0.98 L min$^{-1}$) and used as plasma gas (15 L min$^{-1}$) (PerkinElmer, Waltham, MA, USA). ICP-MS standards, mix 10 ppm, by Inorganic Ventures (Christiansburg, VA, USA) was used for calibration of the analytical equipment.

For POPs analysis, a 500 mL sample aliquot was spiked with internal standards for the PCBs and PAHs analysis ($^{13}$C-PCB 28, $^{13}$C-PCB 180, naphthalene-d8, benzo[a]anthracene-d12; Supelco, USA). A two-stage liquid–liquid extraction with dichloromethane (Sigma Aldrich, USA) was then carried out. After each addition of 15 mL dichloromethane, the samples were shaken for 30 min, and the extracting solvent was transferred into a glass vial. Lastly, the extracts were evaporated to a volume of approximately 300 μL under a gentle stream of nitrogen (99.8% purity; AGA, Poland).

The determination of PCBs and PAHs was carried out with an Agilent Technologies 5975C gas chromatograph coupled with an Agilent Technologies 7890A mass spectrometer. The analytical column ZB5-MS5 (5% phenyl + 95% dimethylpolysiloxane, 30 m × 0.25 mm × 0.25 μm) provided the separation of analytes, with helium as mobile phase, at a flow rate of 1.3 mL min$^{-1}$. The temperature program ramped as follows: initially 40 °C, increase from 40 to 120 °C at 40 °C min$^{-1}$, then 120 up to 280 °C at 5 °C min$^{-1}$, hold for 17 min (PAHs) or 5 min (PCBs). The injection volume selected for all analyses was 2 μL, with splitless injection mode. The mass spectrometer was operated in the selected ion-monitoring

(SIM) mode. As standard solutions, a mixture of 7 PCBs [PCB 28, PCB 52, PCB 138, PCB 153, PCB 101, PCB 118, PCB 180] in isooctane and a mixture of 16 PAHs (acenaphthylene, acenaphthene, anthracene, benzo[a]anthracene, benzo[a]pyrene, benzo[b]fluoranthene, benzo[k]fluoranthene, benzo[ghi]perylene, chrysene, dibenzo[a,h]anthracene, fluoranthene, fluorene, indeno[1,2,3-cd]pyrene, naphthalene, phenanthrene, pyrene) were used, supplied by Restek Corporation USA. The mass to ion ratios monitored are presented in Table S2 (Supplementary Information: water-673537-supplementary). Prior to sample analysis, column performance was checked with relevant standards, as were peak height and resolution. Further quality checks included a solvent blank, a standard solution mixture and a procedural blank, run in each sequence of samples to test for contamination, peak identification and quantification. Thus, the background concentrations, resulting from reagent and analytical vessel quality, were eliminated from final quantitation results. Average recoveries of validation standards were in the range of 70%–85%, while internal standard recoveries amounted to 80%, 85%, 82% and 80%, for naphthalene-d8, benzo[a]anthracene-d12, $^{13}$C-PCB 28, and $^{13}$C-PCB 180, respectively.

Surfactants were determined using 6300 Visible Spectrophotometers, with a set of cells of 10 mm light path length (Jenway, Staffordshire, UK). Dodecane-1-sulphonic acid sodium salt, p-tert-octylphenoxypolyethoxyethanol and *N*-cetyl-*N*,*N*,*N*-trimethylammonium bromide were applied as standard reagents for calibration and spiking (supplied by Merck KGaA, Darmstadt, Germany, or Sigma Aldrich, St. Louis, MO, USA). Formaldehyde and phenols were also determined spectrophotometrically (PHARO Spektrophotometer 100 and 300, MERCK, Søborg, Denmark), using ready-made test kits by Merck (formaldehyde: Test Spectroquant®nr (Merck, Darmstadt, Germany), 1.14678.0001, λ = 495 nm, phenols: Test Spectroquant®nr (Merck, Darmstadt, Germany), 1.00856.0001, λ = 585 nm).

The analysis of SCFAs was performed with ion chromatography DXICS3000 system (Dionex Corporation, Sunnyvale, CA, USA), with UV-Vis detection, fitted with an Acclaim$^{TM}$ Organic Acid column (5 μm, 4.0 × 250mm, Dionex Bonded Silica Products). The mobile phase was $CH_3SO_3H$, 2.5 mM, in water, at a flow rate of 6 mL min$^{-1}$. Injection volume was set to 5 μL. Calibration curves were prepared with organic acid standard solutions at a concentration of 100 mg L$^{-1}$ (TRACECERT, Fluka Analytical, USA).

TOC was determined using catalytic oxidation with oxygen at 680 °C, with non-dispersive infrared spectroscopy as detection method, on a Total Organic Carbon Analyser TOC-VCSH/CSN (Shimadzu). As a calibration standard, potassium hydrogen phthalate, $C_6H_4(COOH)$, FW204.23, purity 99.9% (Kanto Chemical Co., Tokyo, Japan) was applied. All data were subject to strict quality control procedures, as described in Supplementary Information (Table S1 in water-673537-supplementary).

*2.4. Statistical Procedures*

Before statistical analysis (i.e., in the dataset provided in Supplementary Information data file: water-673537-supplementary_new.csv), all values below the limit of detection (LOD) were replaced with half of its value (however, for the calculation of the sums of PAHs and PCBs, the values below LOD were omitted). The basic statistical tests were calculated using STATISTICA 13.1 software (by Statsoft). Kruskal–Wallis ANOVA was used instead of parametric ANOVA due to multiple variables exhibiting non-normal distribution. An additional advantage in using a rank-based procedure, such as Kruskal–Wallis ANOVA, is the fact that the data censoring procedure of assigning ½ LOD has no influence on its result.

The principal component analysis (PCA) was computed in R (version 3.4.4.) with the prcomp package. Prior to analysis, each variable except pH has been log-transformed to bring their distribution closer to normal. The number of 4 PCs explaining a significant portion of total variation in the dataset was picked based on a scree plot. We performed PCA both with surfactant concentrations and without them and their inclusion only changed the result marginally, while their exclusion allowed to plot more data records, which we chose as the preferred option. PAHs, PCBs, and SCFAs have not been plotted in the PCA due to a much smaller number of data points than in the case of other variables.

A sensitivity analysis was also performed to explore the influence of the data censoring on the PCA plots (by replacing <LOD values with a value close to zero or with 1 LOD). The resulting plots have been used as a cross-check validation for the conclusions drawn in the paper (see Figures S1 and S2 in Supplementary Information: water-673537-supplementary).

## 3. Results

### 3.1. EC and pH

The basic characteristics of the freshwater samples collected can be described with their electrical conductivity (EC) and pH (Figure 2). The EC and pH values ranged from 37 to 609 μS cm$^{-1}$ and from 4.28 to 8.21, respectively. The high variability of chemical properties experienced in the area is confirmed by the Kruskal–Wallis ANOVA results, which yield statistically significant differences between the following pairs of samples:

- for pH: Fuglebekken stream—Revdalen lake; Fuglebekken stream—Rålstranda lake; Revdalen lake—stagnant water on the Hans glacier moraine (lake or puddle); Rålstranda lake—all samples from the Hans glacier moraine;

- for EC: Fuglebekken stream—Fuglebekken lake; Fuglebekken stream—Revdalen lake; Fuglebekken lake—all Revdalen sample categories; all Revdalen sample categories—stagnant water on the Hans glacier moraine (lake or puddle).

The highest EC and pH values were both characteristic for the moraine site, where shallow lakes and streams are in direct contact with crushed rock. Shallow lakes located at the seaward plains, but also the stream Fuglebekken, all showed higher ECs than the Revelva system waters.

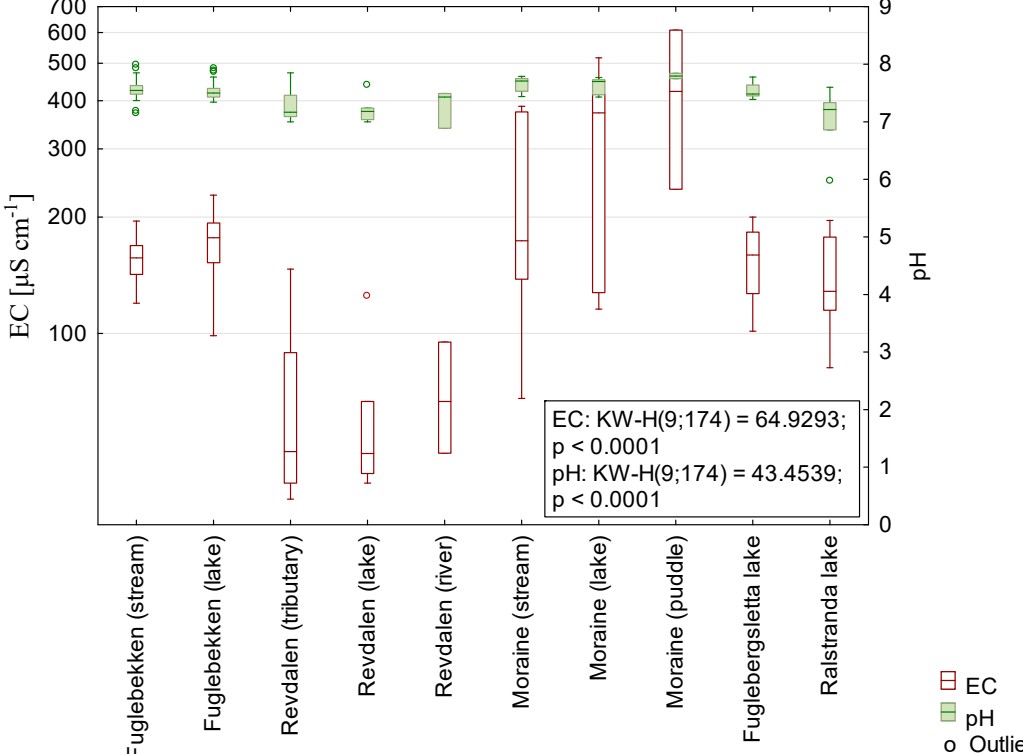

**Figure 2.** A box-and-whisker plot of the basic physicochemical parameters of the surface water samples collected in the Hornsund area: electrical conductivity (EC, in logarithmic scale) and reaction (pH). The box spans from first ($Q_1$) to the third quartile ($Q_3$), with median ($Q_2$) as a line inside. The whiskers show the range of results except outliers, which are defined as values further than 1.5 box span ($Q_3 - Q_1$) above $Q_3$ or below $Q_1$.

### 3.2. Metals and Metalloids

Metals and metalloids have shown significantly varied concentrations across the water bodies studied in the Hornsund area (Figure 3). Among the 12 elements which showed at least n = 50 values exceeding detection limits (out of n = 136), i.e., Ba, Al, As, Cr, Cu, Li, Mo, Pb, Rb, Sb, Se, Sr, Th, U, V and Zn, only Pb did not show significant differences in concentrations between water bodies (Kruskal–Wallis ANOVA results: all other listed elements have shown $p < 0.0001$). Pairwise comparisons of the regional clusters of water bodies are summarised in Table 1. The most distinct chemical composition (signified by multiple significant pairwise differences in metal and metalloid concentrations) was found in lakes: Fuglebekken, Revdalen and those located on the Hansbreen moraine, as well as in the Fuglebekken stream. Among the elements, numerous significant contrasts in surface water concentrations were found for Al, Ba, Cr, Li, Rb, Sr, U and V.

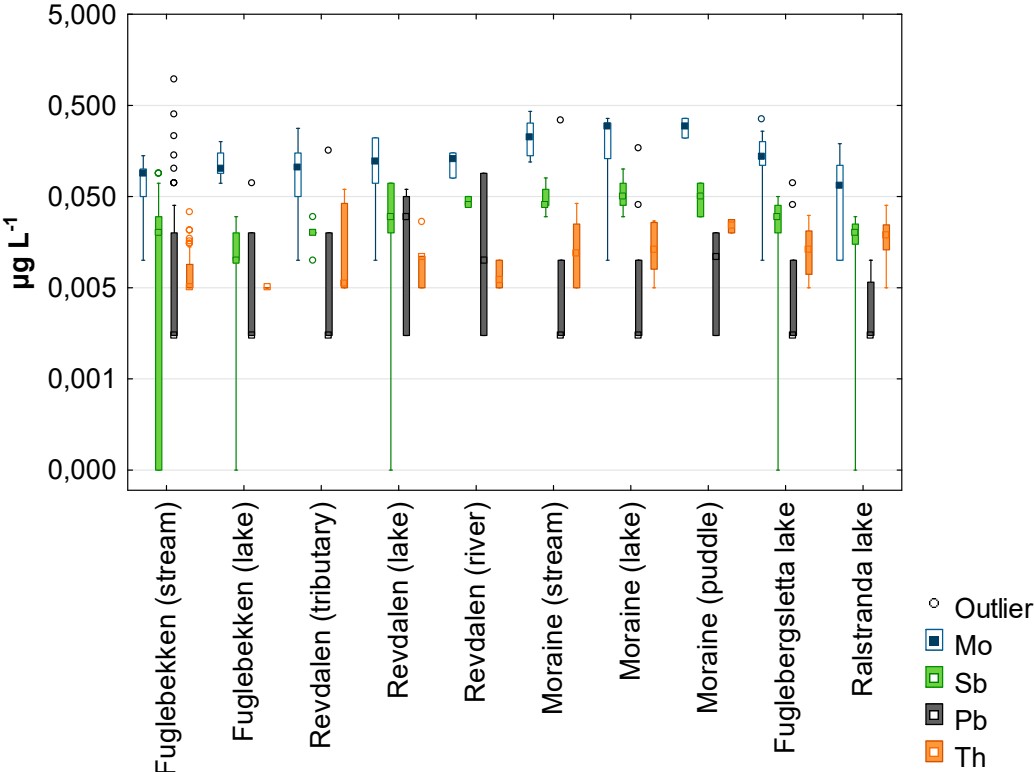

**Figure 3.** *Cont.*

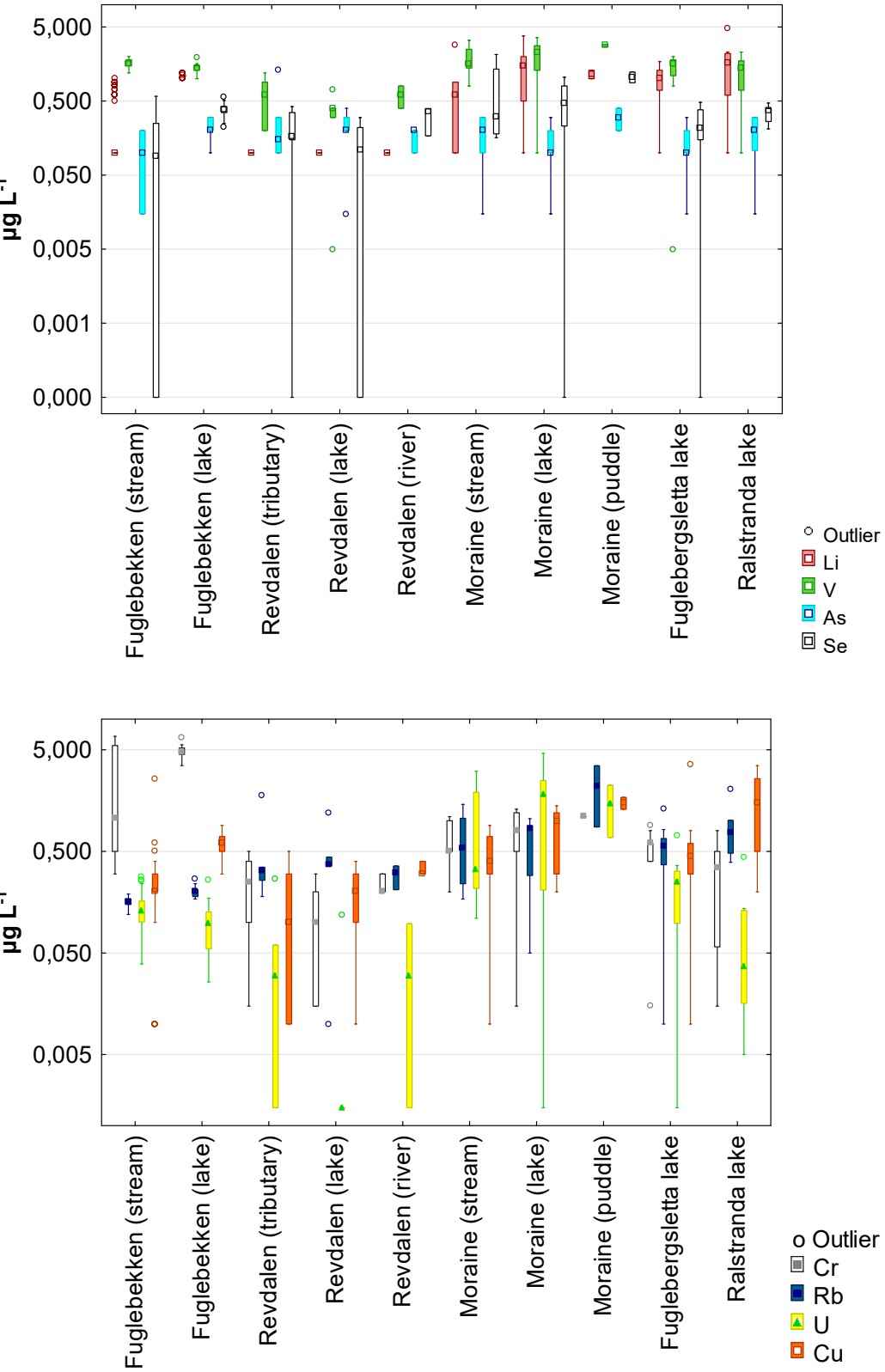

**Figure 3.** *Cont.*

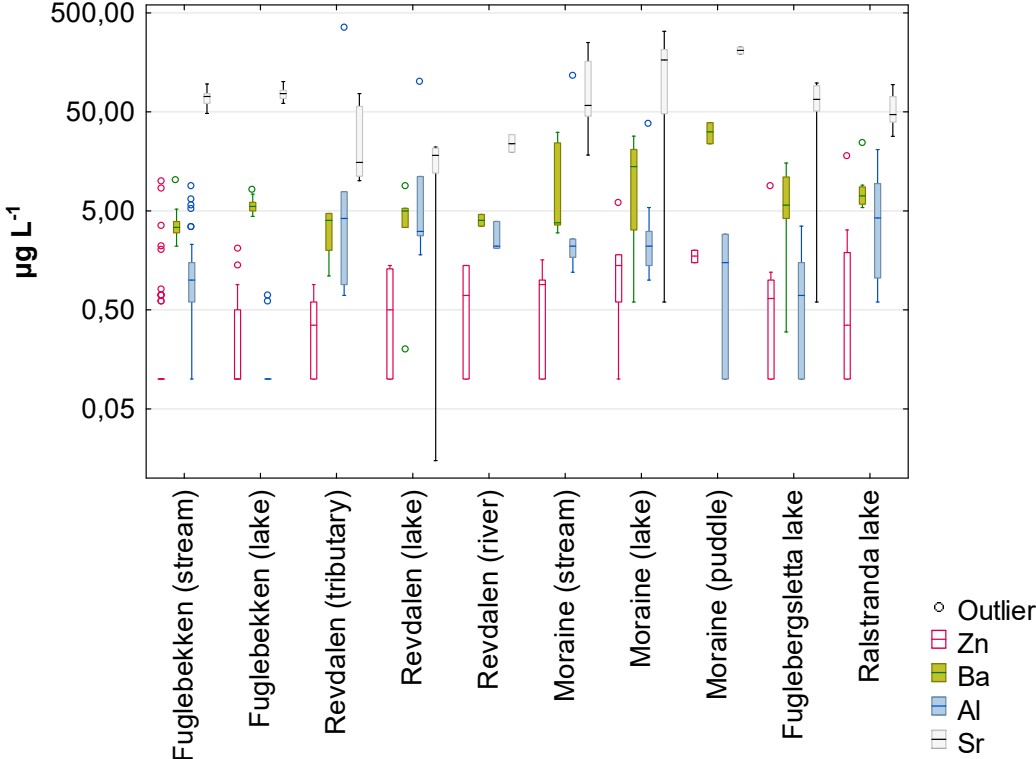

**Figure 3.** A box-and-whisker plot of the metals and metalloids concentrations measured in the freshwater environments in the Hornsund area, divided according to their concentration range into four graphs. All Y-axes are logarithmic scale. The box spans the interquartile range ($Q_1 - Q_3$), with median ($Q_2$) as a marker inside the box. Whiskers show the range of results except outliers, which are defined as values further than 1.5 interquartile range above $Q_3$ or below $Q_1$.

**Table 1.** Statistically significant differences between water bodies in the Hornsund area with respect to metal and metalloid concentrations (according to Kruskal–Wallis ANOVA, $p < 0.05$). Pb not listed due to lack of significant differences between any of the water bodies.

| Pairwise Comparison | Al | As | Ba | Cr | Cu | Li | Mo | Rb | Se | Sb | Sr | Th | U | V | Zn |
|---|---|---|---|---|---|---|---|---|---|---|---|---|---|---|---|
| Fuglebekken (stream), **FS** | FL RL | FL | FL ML FLL RLL | RT RL RLL | FL ML RLL | FL ML FLL RLL | MS ML | FL RT RL MS ML MP FLL RLL | FL ML | ML | RL | - | RL | FL RT RL | ML |
| Fuglebekken (lake), **FL** | FS RT RL RR MS ML RLL | FS | FS | RT RL RR FLL RLL | FS RT | FS RT RL | - | FS | FS RL | MS ML | RT RL | ML FLL RLL | ML | FS | ML |
| Revdalen (lake), **RL** | FS FL | - | - | FS FL | - | FL ML FLL | - | FS | FL | - | FS FL ML MP | - | FS MS ML MP FLL | FS ML MP | - |
| Revdalen (river), **RR** | FL | - | - | FL | - | - | - | - | - | - | - | - | ML | MP | - |

**Table 1.** *Cont.*

| Pairwise Comparison | Al | As | Ba | Cr | Cu | Li | Mo | Rb | Se | Sb | Sr | Th | U | V | Zn |
|---|---|---|---|---|---|---|---|---|---|---|---|---|---|---|---|
| Moraine (stream), **MS** | FL | - | - | - | - | - | FS | FS | - | FL | - | - | RL | - | - |
| Moraine (lake), **ML** | FL | - | FS | - | FS RT | FS RT RL | FS RLL | FS | FS | FS FL | RL | FL | FL RT RL RR RLL | RT RL | FS FL |
| Moraine (puddle), **MP** | - | - | - | - | - | - | - | FS | - | - | RT RL | - | RL | RT RL RR | - |
| Fuglebergsletta lake, **FLL** | - | - | FS | FL | - | FS | - | FS | - | - | - | FL | RL | - | - |
| Rålstranda lake, **RLL** | FL | - | FS | FS FL | FS RT | FS RL | ML | FS | - | - | - | FL | ML | - | - |

### 3.3. TOC, Formaldehyde and Phenols

Summary parameters of TOC and the sum of phenols were also measured in multiple samples from the area, as was formaldehyde concentration (Figure 4). All these parameters showed significant differences between the studied objects (Kruskal–Wallis ANOVA, $p < 0.006$). However, in pairwise comparisons, the environments differing significantly ($p < 0.05$) were found only for the sum of phenols (Fuglebekken lake, with high sum of phenols values, diverged from Fuglebekken stream, the Revvatnet lake, and moraine lakes) and TOC (moraine lakes, with the lowest TOC values, being distinctive from Fuglebekken stream, Fuglebekken lake and Rålstranda lakes). Some extreme outliers were found, especially in TOC concentrations in the Fuglebekken stream.

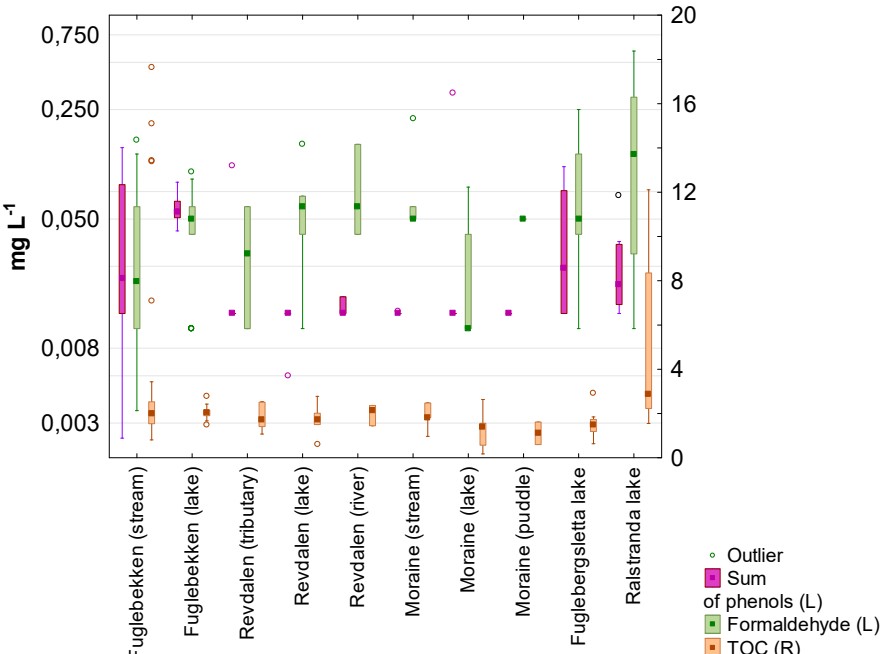

**Figure 4.** Formaldehyde, phenols and total organic carbon (TOC) concentrations measured in the freshwater from lakes and streams in the Hornsund area. (R) or (L) signifies which Y-axis (right or left) the particular parameter is plotted upon (left Y-axis in logarithmic scale). Box = results range from first ($Q_1$) to the third quartile ($Q_3$); filled marker inside the box = median ($Q_2$); whiskers = range of results except outliers; outlier = above $Q_3 + 1.5 \times (Q_3 - Q_1)$ or below $Q_1 - 1.5 \times (Q_3 - Q_1)$.

### 3.4. PAHs and PCBs

With respect to the sum of PAHs concentrations, no significant differences were found across the range of studied surface water objects (Figure 5; Kruskal–Wallis ANOVA chi-square = 10.48; $p = 0.163$). The main contributor to the analyzed sum of PAHs was naphthalene, with the maximum concentration found in Fuglebekken stream at 3950 ng $L^{-1}$, while its minimum concentrations were below LOD. Each of the measured compounds has sometimes been found at extremely high concentrations if compared to the general chemical composition of its water body class (i.e., exceeding three interquartile ranges above $Q_3$), and this was characteristic for Fuglebekken stream and Fuglebergsletta lakes, which were the water bodies located closest to the Polish Polar Station and thus perhaps exposed to emissions from power generator or other local combustion processes (e.g., exhaust from snowmobiles in the spring or amphibia transporters in the summer). On the other hand, the broadly uniform concentrations across most of the study samples in the area suggests a background level of PAHs pollution, disconnected from local human activities in the area, unless they uniformly covered all the studied areas. The relatively high values of PAHs concentrations, e.g., in the Rålstranda lake sample, one of the more distant sites from the PSP Hornsund, highlight the need to further investigate such locations.

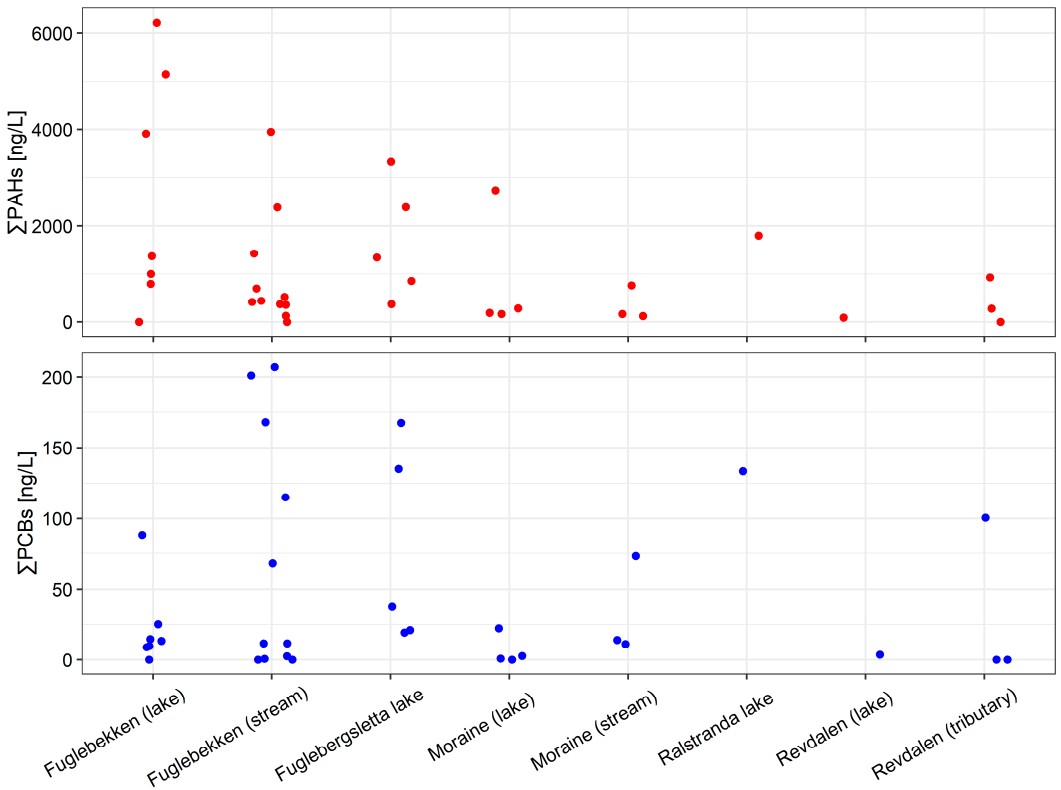

**Figure 5.** A plot of the sums of polycyclic aromatic hydrocarbons (PAHs) and polychlorinated biphenyls (PCBs) measured in the surface waters of Hornsund area (n = 35).

The sum of the measured 6 PCBs (Dutch 7 except PCB-52; Figure 5; Table S3 in water-673537-supplementary; Supplementary Information data file: water-673537-supplementary_new.csv) showed no significant concentration differences between the studied water body groups (Kruskal–Wallis ANOVA, chi-square = 8.06, $p = 0.327$). Again, a relatively high concentration of several PCBs was found in the Rålstranda lake sample. A PCB congener of concern in the whole area was PCB-153, which was the most frequently detected (n = 27 values above LOD, among n = 35 samples tested for PCB concentrations) and which exhibited an order of magnitude higher maximum concentrations than the other measured compounds from this group.

### 3.5. Short-Chain Fatty Acids (SCFAs)

Among organic compounds, short-chain fatty acids (SCFAs) were determined in n = 32 samples (Figure 6). Oxalate and propionate did not exceed LOD in any of the tested samples (valerate and isovalerate did exceed LOD only in one sample), while the highest concentrations among the determined compounds were found for acetate (especially in Rålstranda and Fuglebekken lakes). Despite some variety in the obtained concentrations, most differences between the sampled freshwater media were not statistically significant (Kruskal–Wallis ANOVA with $p < 0.05$ was only found for acetate, and pairwise comparisons for acetate did not yield any statistically significant results).

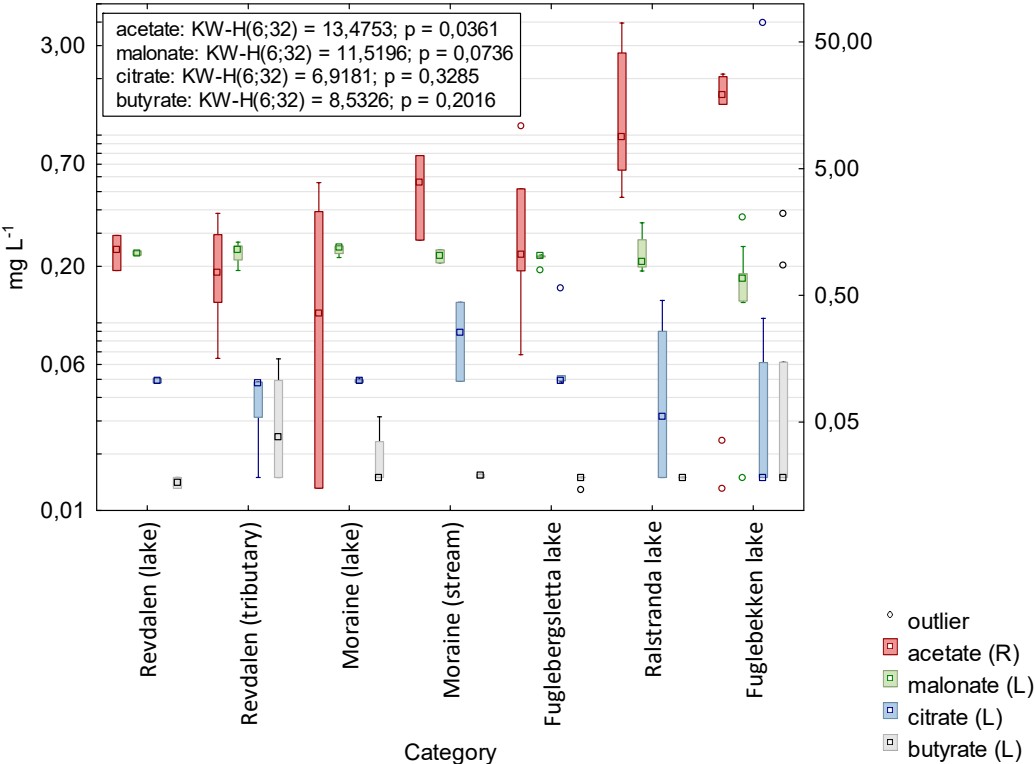

**Figure 6.** Selected short-chain fatty acids (SCFAs) concentrations, which exceeded detection limits in at least n = 9 samples, from lakes and streams in the Hornsund area. (R) or (L) signifies which Y-axis (right or left) the particular parameter is plotted upon. Box = results between first ($Q_1$) and third quartile ($Q_3$); empty square = median ($Q_2$); whiskers = range of non-outlier results; outlier = a value further than 1.5 box span ($Q_3 - Q_1$) beyond $Q_3$ or $Q_1$.

### 3.6. Surfactants

Surfactant concentrations (Figure 7) differed between the studied water bodies (Kruskal-Wallis ANOVA, $p < 0.05$) for anionic and cationic detergent types. In pairwise comparisons, such effect was only found for anionic surfactants between moraine lakes and Fuglebekken stream as well as Fuglebergsletta lakes. Furthermore, the highest concentration of any single surfactant class was found for anionic detergents in a moraine lake (closest to the PSP Hornsund). However, stagnant waters were not necessarily the most abundant in detergents. Nonionic surfactants were most frequently not detected (n = 98 out of n = 106 results were below LOD), perhaps due to the higher detection limit of the applied test for this class of compounds, and their presence was only confirmed in single samples from Fuglebekken stream, Revdalen lake, river and its tributaries, in a Hans glacier moraine stream, and in a Fuglebergsletta lake.

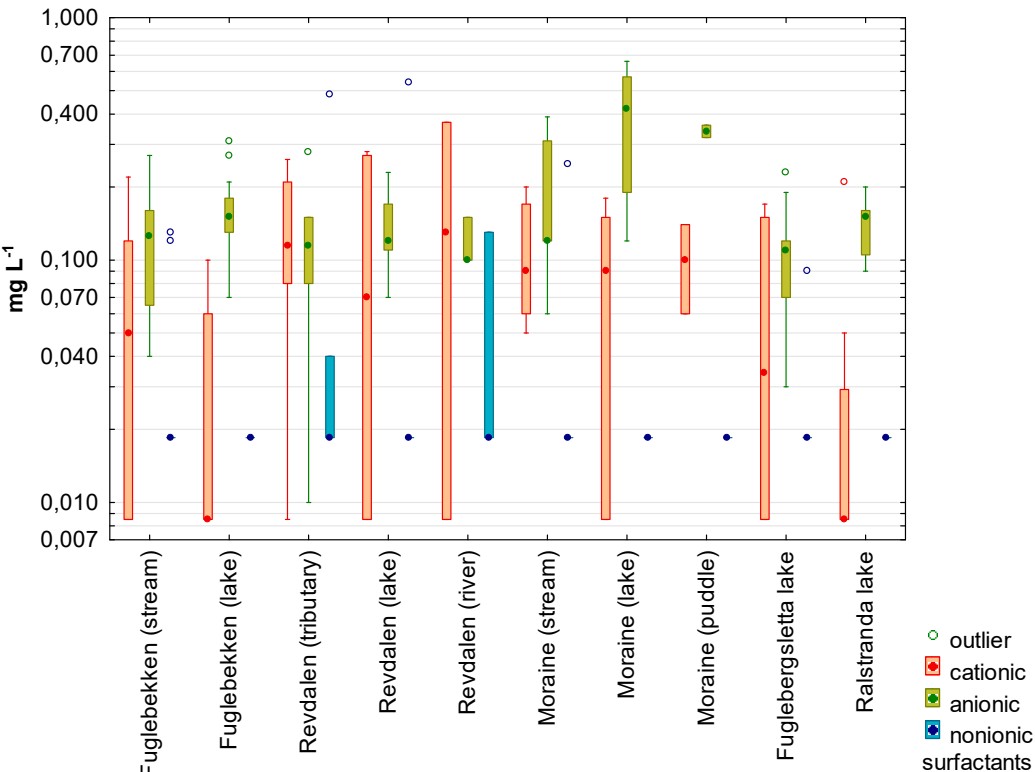

**Figure 7.** Concentrations of three groups of surfactants in the surface waters from Hornsund area. Box = results between first ($Q_1$) and third quartile ($Q_3$); filled circle = median ($Q_2$); whiskers = range of non-outlier results; outlier (empty circle) = a value further than 1.5 box span ($Q_3 - Q_1$) beyond $Q_3$ or $Q_1$.

### 3.7. Principal Component Analysis

To distinguish the influence of the local anthropogenic emissions from the geological background and other natural factors, we have conducted principal component analysis (Figure 8; see also PCA scores file: Supplementary_PCA_scores) with the concentrations of metals and metalloids, EC and pH of the samples, their TOC, HCHO and sum of phenols concentrations, and their distance from the PSP and the sea. The latter was measured as a shortest distance to the shore, except in the deep valley Revdalen, where it was measured along the valley axis. We have highlighted the different sampling areas and types of water bodies in the resulting plots.

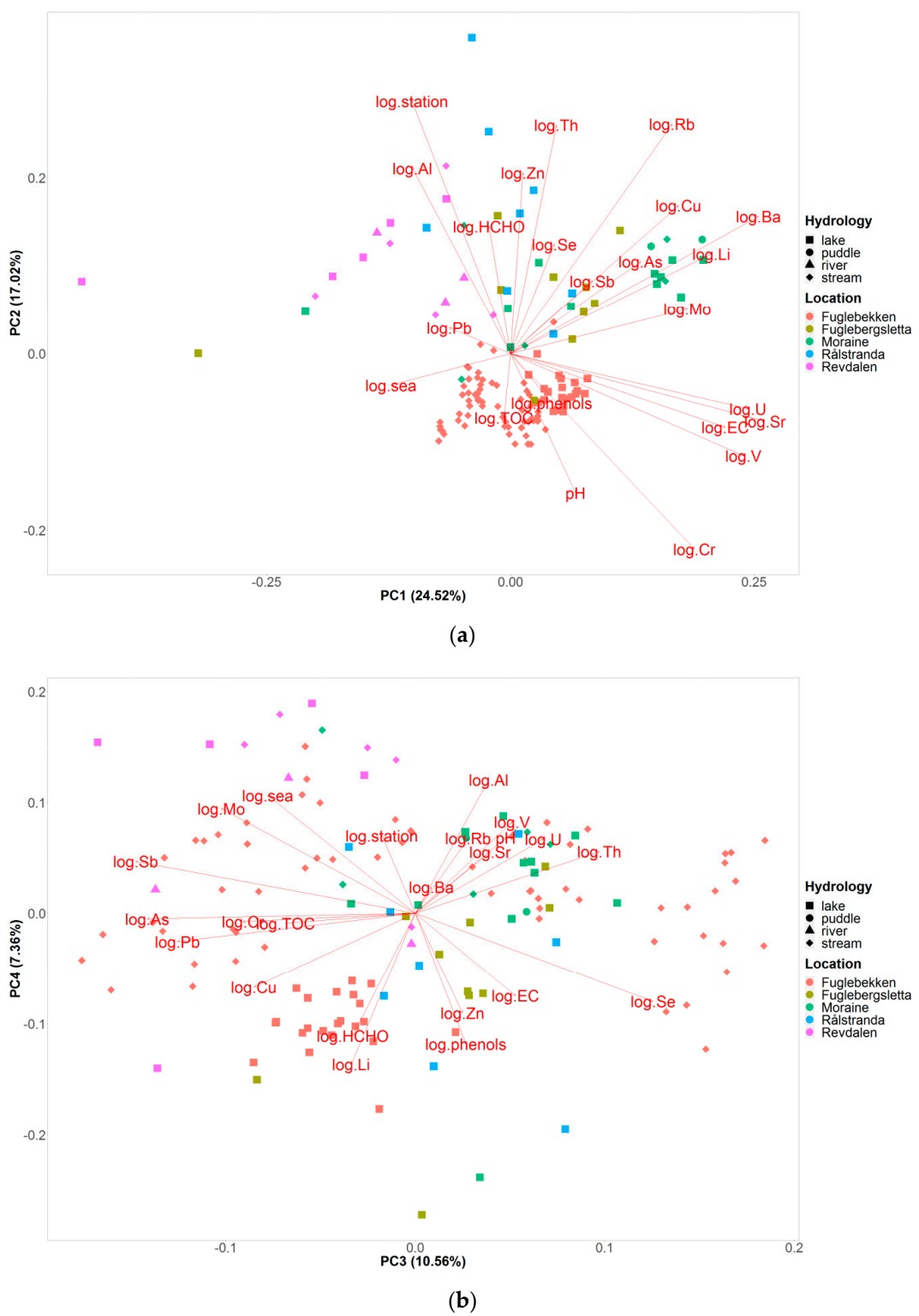

**Figure 8.** Principal component analysis (PCA) of the dataset consisting of EC, pH, and the concentrations of metals and metalloids, surfactants, TOC, formaldehyde and the sum of phenols, with distances from the Polish Polar Station (PSP) Hornsund and from the shoreline included. The variables are plotted in a space defined by: (**a**) pricinpal components (PCs) 1 and 2; (**b**) PCs 3 and 4. Color and shape of the points show their location and water body type.

From PCA, it can be seen that the regional differences in surface water composition in this area are more pronounced than those related to the type of water body, including whether the water is flowing or stagnant. This lends more credit to the use of this mixed dataset to trace the influence of various local chemical factors. Only in the case of Fuglebekken stream and lake, the bigger datasets related to those objects were clearly divided in the PCA plot, yet they were still relatively close in chemical composition.

Principal component (PC) 1, which explained 24.52% variability of the dataset (Figure 8a), differentiated the various locations represented in this study. However, it merged Fuglebekken, Fuglebergsletta and Rålstranda into one category, which may be both due to their proximity to the shoreline and their similar underlying geological substratum (formations Ariekammen and Janusfjellet, which occur in a belt alongside the shore of the Hornsund fjord). The PC1 correlated strongly with EC, and the concentrations of V, Sr, U, Li, Ba, Mo, As, Cu and, negatively, with the distance from the sea. Of these elements, V, Sr, U, Li, Ba were also confirmed by Kruskal-Wallis ANOVA to effectively differentiate local water body groups.

The PC2 (17.02% variability explained, Figure 8a) correlated with the distance from the station, as well as the concentrations of Al, Zn, Th, Cu and Rb, and (negatively) with the concentration of Cr. Of these, Al, Rb and Cr were confirmed by Kruskal-Wallis ANOVA to exhibit multiple significant spatial differences in the area. In the space defined by PCs 3 and 4 (explaining 10.56% and 7.36% variability in the dataset, respectively; Figure 8b), EC shows a negative correlation with distance from the sea, and Se concentration followed the same pattern as EC in that space. On the other hand, Mo, Sb, As and Pb concentrations exhibited an opposite behavior there.

## 4. Discussion

Arctic catchments are sensitive to external pollution, and yet what is frequently considered external contamination may also originate from local sources and natural processes in the area. As a result, a complex chemical composition of waters is frequently difficult to assign to the actual origins. In this work, we analyze the chemical composition of surface waters in the catchments surrounding the Polish Polar Station in Svalbard with a spatial focus, in connection to the local sources of such impurities. As a result, a more comprehensive picture of the local geochemistry is presented, which includes geological composition of the substratum, and the close proximity of both the research station and the sea surface (Hornsund fjord). This needs to be also placed in the context of frequently described long-range pollution transport into the Arctic [2,28–32], which includes both industrial sources and an increasing influence of biomass burning events [33].

The chemical composition of local waters has been described so far with respect to trace elements concentration [10,11,13–15], some of these publications also included inorganic ions, surfactant and total organic carbon (TOC) concentrations, as well as pH and electrical conductivity (EC). Contamination with PAHs and PCBs was described by [9,12,16]. In [16], 30 water samples collected in 2010 in the area were described, with the local concentrations of 16 PAHs ranging from <LOD to 6212 ng L$^{-1}$, with a predomination of naphthalene among the detected compounds. Among the 7 Dutch PCBs (with total concentrations ranging from <LOD to 273 ng L$^{-1}$), the highest concentrations were found for PCB-153.

In this study, we provide a more comprehensive picture of the chemical composition of surface waters in the area, with regional differences and a wider scope of the analyzed compounds. Our attention is drawn especially to the local sources of impurities, such as the geological substratum, marine aerosol and the only human settlement in the area—the Polish Polar Station. Since the area is geologically varied and underlain mainly by metamorphic rocks (with newer ore-bearing veins) [22,24,26,34], the substratum is a likely source of multiple metals and metalloids in the surface waters. Furthermore, the ubiquitous influence of marine aerosol has been described in local precipitation waters (e.g., by [35]).

The PCA performed can be used to distinguish important sources, correlating with a cluster of chemical variables (Figure 8). For all the elements correlating with PC1 (Figure 8a), except As and Mo, (i.e., for V, Sr, U, Li, Ba, and Cu), geological sources are known either in Fuglebekken or Revelva

catchment, or both [10,15]. Especially Ba distinguished, by its high concentrations, selected samples from the Hans moraine, with an additional influence of Li, As, Mo, Cu, Zn and Th. Since it is an area where finely crushed rocks have been brought from further up in the Hans glacier basin, it is worth exploring the geological sources of these elements in rock formations not directly adjacent to the moraine. In particular, this applies to the formations Slyngfjellet from Sofiebogen Group, and formations Bergskardet, Bergnova, Tonedalen from the Deilegga Group, all of which underlie the higher parts on the western side of Hans glacier. These formations consist of, in the above-mentioned sequence, of conglomerate, quartzite, a phyllite-slate complex of thin dolostone and pyritiferous shale (with secondary gypsum), and quartzite conglomerate with dolostone. Gangue minerals and ores found in veins of the Deilegga Group are known to contain Al, Mn, Zn, Li, Cu, Pb, As and Sb; As was found in those as arsenopyrite and as an admixture in pyrite, chalcopyrite, galena and sphalerite [34]. Metamorphic dolostones have also been observed to contain relatively high admixtures of barium [36]. Furthermore, Cu was found in ore-bearing (chalcopyrite) veins by Wojciechowski [26] on the moraine itself. The remaining element in the above considerations is molybdenum, for which we cannot supply a geological source in the area. However, it is the most plentiful transition metal in open seawater [37] and some variability in its concentration can be explained by the proximity to the sea shore (Figure 8a), therefore a likely (but not singular) source of it is either the abundant sea spray or melting permafrost from the uplifted marine terrace upon which some of the lakes are located.

On the other hand, rain and snow may also be sources of metals and metalloids in the surface waters in the area [10]. Long-range transport may bring As, V or Cr (since the concentration of these elements depends strongly on the air mass trajectory leading to a precipitation event [10]). The local winds are likely to bring Mn, U and Ba into rain or snowfall from the east and north-east, and B, As, Rb, Se, Sr and Li from the south-west [10]. However, in the case of surface waters, especially the small stagnant lakes, the presence of local rock sources in sediment seems a much more plausible source of metals and metalloids than precipitation, unless the latter exhibited an unusually high concentration of a certain element. In the year of field investigation, Mn, Sr and Zn were found at particularly high maximum concentrations in a single precipitation event (12.5, 32.6 and 137.2 $\mu$g L$^{-1}$, respectively—see Table S4 in Supplementary Information: water-673537-supplementary, for more precipitation data). Nevertheless, the Mn and Zn concentrations exceeded the LOD only in a limited amount of samples (in n = 52 for Zn and n = 22 for Mn). Furthermore, a precipitation event with a high elemental concentration should affect the whole area uniformly, so its impact should only reflect the dilution according to the size of water body (which should produce a significant difference between the Revdalen lake and the other, much smaller water bodies). This has only occurred in the case of Sr (Figure 3, Table 1).

Chromium concentration, negatively correlated with both PC2 and the distance from the PPS (Figure 8a), could come from, for example, diesel generator exhaust [38,39]. However, the same source is likely to release also Zn or Pb, or even As, for which a marked pattern of this kind did not occur. The lack of it indicates that Zn, Pb and As from local geological sources mask their possible anthropogenic emissions in the area. Especially zinc can show elevated concentrations in upper Revdalen from sphalerite veins found in that location [26]. Al, Rb, Cu and Th concentrations may also increase further from the station due to a stronger influence of rock dust upon the samples collected in upper Revdalen, where frost-weathering slopes are exposed (and the local rocks contain those elements [21,22,26]).

Interestingly, contrary to the suggestions of Wojtuń et al. [20] that there is an incinerator impact in Co, Ni and Li concentrations in mosses, these elements have not been particularly frequently detected in the surface waters (Supplementary Information data file: water-673537-supplementary_new.csv). Furthermore, their concentrations peaked at locations further away from the station. Of these, only Li is pictured in the PCA plots and it does not show a relationship with distance from the station.

In the space defined by PCs 3 and 4 (Figure 8b), the opposite relationship of EC and Se to the distance from the sea could represent the marine influence. While the highest EC values were found

in the moraine samples (likely as a result of contact with crushed rock), there was generally higher EC in all samples located at the seaward plains. These have been formed (at least in part) as a raised marine terrace, and thus there may be saline groundwater increasing the EC in the lakes. Selenium concentration following the same pattern as EC in that space of the PCA aligns with the marine explanation of this factor, since among various sources of Se in the environment, sea spray is mentioned among the important ones [40]. While there may be admixtures of Se in the local mineral veins (as enrichments in galena, pyrite, pyrrhotite, or in smaller amounts in ankerite and magnetite), these are not influencing the local surface water composition as strongly as rock-derived As or Pb. It seems from the PC3-4 space that Sb and Mo are more likely derived from rock sources than the station or the sea, which should encourage searching for geological sources of molybdenum in the area, despite the lack of direct reports from geological surveys on this.

Due to the smaller subset of samples providing the concentrations of PAHs, PCBs and SCFAs, these chemicals have been excluded from the PCA. Furthermore, when their concentrations were plotted against the distance from the station and from the seashore, it was evident that within their relatively small datasets outliers strongly influenced the direction of a possible relationship, and the values below LOD could also influence the correlation result. Therefore, we tested the interdependence between these chemicals and the distances from station and sea using a rank-based measure (Spearman correlation coefficient). Moreover, only variables with more than half of the dataset consisting of >LOD values were included in this analysis. These variables were: naphthalene, PCB-153 and PCB-180 (as well as the sums of PAHs and PCBs). Of those, only naphthalene exhibited a statistically significant ($p = 0.006$) and negative correlation with the distance from the PSP Hornsund (and not with the distance from the sea). This suggests the station is a local source of this particular compound. Such an impact is plausible due to the use of various petroleum-based fuels in the running of the station, especially power generation with diesel engines. Among the SCFAs, only acetate, malonate and citrate fulfilled the data usability condition, and among those, only malonate exhibited statistically significant ($p = 0.027$) and positive Spearman correlation with the distance from the station (more results of these analyses are provided in Table S7 in Supplementary Information: water-673537-supplementary).

For both PAHs and PCBs, some high outliers occurred more frequently in the vicinity of the polar station and the sea, however, their number was insufficient to show a uniform pattern. Such a patchy distribution of these pollutants may be shaped by multiple phenomena, e.g., (1) the occurrence of episodic point sources related to field trips from the station (by boat and snowmobile), (2) a mixed influence of local contamination and long-range transport of POPs which are more likely deposited at higher altitudes due to the cold condensation phenomenon [41], or (3) remobilization from environmental reservoirs (e.g., from sediment disturbed during a high discharge episode in a stream). The last hypothesis includes the possibility of persisting pollution from the past operations in the area [17,19]. With respect to hypothesis no. 2, the pattern of PCBs concentrations is not consistent with a local influence of the cold condensation effect, since the more volatile PCBs are not showing higher concentrations in the seashore samples. This effect is more likely to shape the large-scale patterns in compound concentrations and thus the increased concentration of PCB-153 in the whole region can be partly attributed to it [42,43]. Another effect may play a role here, too: unlike PCBs with lower chlorination level, the PCB-153 does not undergo intensive volatilization from Arctic lakes in the summer [44]. Unfortunately, the PCB-153 cannot be used as a marker of a single pollution source, since there has been no published accounts of its specific origin. Rather, it is considered a major PCB congener found in most environmental and human matrices which indicates the total PCB level [45]. While at this stage it is not possible to credit a single interpretation of the distribution of the analyzed POPs in this location, the continued research will point to them if focused on the well-represented compounds in connection to their potential emissions characterization.

It should be noted here that some of the high PAHs concentrations at sites distributed at random distances from the Polish Polar Station may be due to the local snowmobile traffic related to the research activity in the area. For comparison, we provide concentration of PAHs in a snow sample

collected on one of the most frequented snowmobile tracks in the area (at the entrance to the Hans glacier moraine crossing) collected in May 2019 (Tables S5 and S6 in Supplementary Information: water-673537-supplementary). The concentration of the sum of PAHs in this sample amounted to 7270 ng $L^{-1}$, and the predominant compounds in it were phenanthrene, anthracene and naphthalene (all at concentrations exceeding 1000 ng $L^{-1}$).

In SCFAs concentrations, the only distinguishable spatial patterns were the higher concentration of acetate in Rålstranda lakes and of butyrate in Revdalen tributaries, which skewed regression patterns strongly enough to indicate significant influence of distance from the station upon acetate concentration and distance from the shore on the concentrations of butyrate. While acetate has been frequently found in arctic snow [46–48] and Drake et al. [49] suggest that melting permafrost may be a ubiquitous source of both acetate and butyrate, it does not clearly explain the encountered spatial pattern of their concentrations.

## 5. Conclusions

Through the statistical analysis of pooled surface water chemical data from the Polish Polar Station area (collected in the summer of 2010), we have found spatial patterns in the concentration of metals and metalloids, some organic compounds and additional chemical parameters. The two most important factors for the elemental composition of these waters appear to be the variety of geological substratum and the proximity of the sea. The long-range transport of pollution likely contributed to a background concentration level of anthropogenic pollutants and not to the experienced differences. The presence of the station is most likely impacting the local concentrations of naphthalene and chromium, but may also play a role in the case of other contaminants, only its influence is masked by the higher variability connected to other sources.

While the data has been collected during the same part of the summer season, which allows for robust spatial comparisons, the temporal representativeness of such data is limited due to hydrological regime changes influencing seasonal characteristics of pollutant concentrations in waters of the area [13,15]. Most anthropogenic substances, such as the PAHs (except naphthalene) and the PCBs, did not show clear spatial patterns, yet the singular location of higher concentrations may indicate episodic and local influence of the human activities in the area. Nevertheless, stronger evidence is needed to define the spatial and temporal extent of such influence. We would recommend a closer examination of the concentration levels of PCB-153 in this environment, which has been found at higher concentrations than any other PCB. On the other hand, studies of metal and metalloid concentrations could detect minor anthropogenic impacts once the geological influence is quantitatively assessed in this area. The locally elevated concentrations of other organic compounds, such as the anionic surfactants, acetate or butyrate, suggest there are some unidentified local sources of those. An investigation into the natural emissions from the plants and soils (including permafrost) would bring new information on the composition of surface waters in the area.

**Supplementary Materials:** The following are available online at http://www.mdpi.com/2073-4441/12/2/496/s1: Supplementary Information data file: *water-673537-supplementary_new.csv*; PCA scores file *Supplementary_PCA_scores.csv*; Supplementary Information text with tables and figures: *water-673537-supplementary.docx*, with the following contents: Table S1. Analytical precision parameters used in this study. Table S2. Ions used in SIM analysis of PAHs and PCBs. Table S3. The summed concentrations of the measured 16 PAHs and 6 PCBs (Dutch 7 less the PCB-52) in the studied water bodies (clustered by landform within which they occur). Table S4. The results of chemical analyses in precipitation samples from the vicinity of the Polish Polar Station Hornsund collected in the summer 2010. Table S5. Analytical parameters for the PAHs determination in the 2019 sample. Table S6. The concentrations of the measured 16 PAHs in a snow sample collected from a frequented snowmobile track in the Polish Polar Station vicinity. Table S7. The Spearman rank correlation coefficients and their significance for the PAH and PCB concentration indicators with the most complete datasets. Statistically significant coefficients have been highlighted in bold font. Figure S1. Principal component analysis (PCA) as in Figure 8, except the <LOD values in the original dataset have been replaced by 10–12. The variables are plotted in a space defined by: (a) pricinpal components (PCs) 1 and 2; (b) PCs 3 and 4. Colour and shape of the points show their location and hydrological object type. Figure S2. Principal component analysis (PCA) as in Figure 8, except the <LOD values in the original dataset have been replaced by the full LOD. The space is defined by: (a) PCs 1 and 2; (b) PCs 3 and 4. Legend as in Figure S1.

**Author Contributions:** Conceptualization, K.K.; methodology, K.K., M.R., F.P., S.C. and Ż.P.; formal analysis, K.K.; investigation, K.K., M.R., S.C. and Ż.P.; resources, M.R., S.C. and Ż.P.; software, K.K., data curation, M.R. and Ż.P.; validation, Ż.P.; writing—original draft preparation, K.K., M.R. and F.P.; writing—review and editing, K.K., M.R., F.P., S.C. and Ż.P.; visualization, K.K. and M.R.; supervision, Ż.P.; project administration, Ż.P.; funding acquisition, K.K. and Ż.P. All authors have read and agreed to the published version of the manuscript.

**Funding:** Preparing the article was partially funded by the National Science Centre of Poland project no. NCN 2017/26/D/ST10/00630.

**Acknowledgments:** Field assistance by the Polish Polar Station Hornsund staff is appreciated, as are laboratory analyses by Katarzyna Cichała-Kamrowska, Ewa Olkowska and Katarzyna Kozak. Michał Ciepły offered valuable comments on cartographical visualisation. David Quiroga is thanked for coding advice for the statistical analysis in R.

**Conflicts of Interest:** The authors declare no conflict of interest. The funders had no role in the design of the study; in the collection, analyses, or interpretation of data; in the writing of the manuscript, or in the decision to publish the results.

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
