# Peer review of "Spatial Differences in the Chemical Composition of Surface Water in the Hornsund Fjord Area: A Statistical Analysis with A Focus on Local Pollution Sources"

_water, doi:10.3390/w12020496_

Round 1
Reviewer 1 Report
The manuscript is devoted to the statistical analysis of surface waters sampled or reported close to the Polish Polar Station in Svalbard, Norway.
The main interest of the paper would be im the assessment of pollution sources in this remote region although the results presented have a limited scope.
One of the problems of the paper is that it uses data from several earlier sources... what is not a problem but address the interested reader to track by themselves what has been previsouly done and is not explicitely included in the paper. Worth mentioning is the absence of hydrogeological information and their contribution to the chemical characteristics of the studied waters. On the other hand, the authors provide with a very comprehensive description of analytical methods, LODs, LOQs, etc, (in Table 1). In the reviewer's opinion, this information should not be and it is advised to derive it to an annex. Conversely, the saved space should be invested in providing a more comprehensive background of the data used (what it is and what it is not available, in addition to what the authors include as new analyses in their manuscript).
Although many things require an in-depth re-appraissal, the reviewer is specially concerned with the discussion that the authors present. It is worth to indicate that the pollutant-source issue is barely discussed (in contradiction with the title of the paper). Furthermore certain observations are not relevant. For instance: the WHO limits for surface waters are intended for human consumption while the authors should focus in the ecological impacts (of locally sourced... that can be ammended or regionally-sourced... with far more complex impacts and not easy ammendment). The comment about "increased risk of cancer" (page 16, line 359) is clearly non-relevant unless a context is provided. Finally, the conceptuallization of the geological substrate as local pollution source makes no sense in the context of a natural area without mining pressures. Authors should consider the establishment of reference natural hydrochemical background concentrations prior to state that there is pollution in the system.
Specific comments follow:
In page 7, lines 207-208:The authors indicate that values below LOD arereplaced with 1/2 of the LOD value. This may be a common approach in many works for data censoring but the reviewre cannot back this as an acceptable statistical practice. The bottom end is: by doing that, are the authors overestimating or infraestimating certain concentrations? Any conclusion derived from the lack of satisfactory answer to this question render conclusions questionable and unsupported.
In page 8, lines 227 and 231 (and elsewhere in the text). Author mention "stagnant water". What is their residence time? Are these water masses influenced by ground waters?
Authors do not provide any information ambut precipitation in the study period (amount, distribution and hydrochemical characteristics). Due to the relevance in transferring a fingerprint to surface waters, authors should provide some information about that.
In figure 2 (and others in the text). The concept of "extreme outlier" has not statistical significance, i.e. if it is an outlier it is an outlier. In whateverthe case the authors shoudl justify that outliers belong (or not) to the same population and if the observed pattern may obey to unsufficient data.
In figure 2:The probability limit is not well written
In page 8, lines 241-242. Replace "hydrological objects" by water masses or something similar
Table 2 is not easy to read and, eventually an oversimplification of connections as it only provides bi-directional non-intuitive relationships. It is suggested to use some sort of "panel plot" for multiple correlation analyses.
There are evident patterns in data distribution for many elements that have not been discussed in the text (e.g. Mo, Sb, U, etc. ) Authors should not stop in the statistical description but to deepen into the significance of patterns.
For all the figures in the text: Simplify legends to avoid unnecesary repetitions
In figure 3. In all the box&whisker plots, avoid the inclussion of censoring-derived data in order to not bias the representation. Do not compute statistical descritors with the censoring criterion indicated as it artificially bias data towards lower values.
Figures 5 and 6 are meaningless. Authors should explore alternative ways to represent their findings (add concentrations of like-compounds to show trends, select representative compounds, etc. On the other hand, what is the source of PCB153? The simple observation of the presence of a compound is not as important as to try to reveal why this compound is present in the system (and its source)
In page 17, line 364. Marine aerosol data is not presented. It is advised to perform a study about the contribution of precipitation to the local surface water studied.
The PCA analysis presented does not include information about the scores obtained. The reported weights in the variance of the different components are nothing what cautions against its representativity and significance.
Author Response
Dear Reviewer,
many thanks for the careful reading and consideration given to our article. In light with your remarks, we have prepared a new version of the manuscript for your perusal, focusing especially on the rigour and readability of our work.
Concerning major issues highlighted by both Reviewers, we have applied the following changes:
The Introduction has been changed to include parts of the more generalised field site description, as well as the literature presenting the wider than local importance of such work undertaken in the vicinity of the Polish Polar Station Hornsund. We maintained the reference to „local sources” for elevated concentration of certain elements and compounds from natural sources (however, they are not named „pollution” directly in text). Although it would have been more accurate to calculate the geochemical background, we have limited information available which would be necessary for this procedure to be conducted rigorously. In Materials and Methods, more information is provided about the novel elements in this work as opposed to the once-published and now pooled data. The details of analytical quality assurance and quality control have been moved to a Supplementary Information In response to concerns as to the interpretation of the principal component analysis (PCA), in the context of the use of values below the detection limit with a censoring procedure of replacing those with ½ LOD, we have performed a sensitivity analysis. It is presented in the Supplementary Information We have also supplied PC scores for the main analysis occurring the article (as a supplementary file in csv format). The discussion of the sources of pollution has been expanded, including information about the potential input from precipitation and snowmobile exhaust. We have also removed the reference to WHO limits as not relevant to the study subject. At the correction stage, we have also noticed a mistake (ommission of a few data points from Fuglebekken lake for SCFAs, PAHs and PCBs).We sincerely apologise for any inconvenience caused by this. Now the whole article has been corrected to include this part of the dataset.Responses to specific comments:
In page 7, lines 207-208: The authors indicate that values below LOD arereplaced with 1/2 of the LOD value. This may be a common approach in many works for data censoring but the reviewre cannot back this as an acceptable statistical practice. The bottom end is: by doing that, are the authors overestimating or infraestimating certain concentrations? Any conclusion derived from the lack of satisfactory answer to this question render conclusions questionable and unsupported.
In general this is one of the valid approaches, and perhaps better would be to replace those with a lower value, e.g. 1/3 of the LOD, since that would reflect better the typically logarithmic distribution of the chemical concentration data. However, in practice, such a difference is probably incosequential for the type of analysis we used. For the sum calculations, we have excluded the values below LOD, so please note these sums are conservative estimates of the pollution level in the area (with the highest values of pollutant concentrations, which are the most important, the resulting difference was negligible). In the case of Kruskal-Wallis ANOVA this is a non-issue. Kruskal-Wallis ANOVA uses rank instead of value, so no matter what we replaced the <LOD with, such values would obtain the same rank. The only analysis for which this could be a problem was PCA, thus we have run a sensitivity analysis for it, which is presented in the Supplementary Information.
In page 8, lines 227 and 231 (and elsewhere in the text). Author mention "stagnant water". What is their residence time? Are these water masses influenced by ground waters?
By stagnant we meant not flowing in a current or stream; we used the term in the hydrological sense, i.e. to encompass lakes and ponds (it also includes wetlands and marshes, which were not studied here). To enhance the available information on the hydrological characteristics of the studied objects, we have also added extra information obtained during the 2010 fieldwork in the Field site section.
Authors do not provide any information ambut precipitation in the study period (amount, distribution and hydrochemical characteristics). Due to the relevance in transferring a fingerprint to surface waters, authors should provide some information about that.
We have collected fragmentary data on the precipitation chemistry in the 2010 season, some have also been published in the previous articles and are quoted in the manuscript (after Kozak et al., 2015). A description of the additional 2010 data has been added to the presented study.
In figure 2 (and others in the text). The concept of "extreme outlier" has not statistical significance, i.e. if it is an outlier it is an outlier. In whateverthe case the authors should justify that outliers belong (or not) to the same population and if the observed pattern may obey to unsufficient data.
In figure 2:The probability limit is not well written
The p-level description has been corrected.
In page 8, lines 241-242. Replace "hydrological objects" by water masses or something similar
Since „water masses” is a term used in oceanography, we have decided to replace "hydrological objects" with „water bodies” instead.
Table 2 is not easy to read and, eventually an oversimplification of connections as it only provides bi-directional non-intuitive relationships. It is suggested to use some sort of "panel plot" for multiple correlation analyses.
Table 2 is based on analysis which highlights whether the geographically (regionally) distinguished groups of water bodies differ significantly in terms of their chemical composition. For the results showing a significant difference in the total data pool, we have run the post-hoc pairwise comparisons to distinguish which groups of objects actually differ from one another and the result of this procedure is the presented table. The matching plot is the box-and-whisker plot in Figure 3. The main reason behind presenting the table is thus to highlight whether there are actual differences worth analysing further.
There are evident patterns in data distribution for many elements that have not been discussed in the text (e.g. Mo, Sb, U, etc. ) Authors should not stop in the statistical description but to deepen into the significance of patterns.
For all the figures in the text: Simplify legends to avoid unnecesary repetitions
Corrected.
In figure 3. In all the box&whisker plots, avoid the inclussion of censoring-derived data in order to not bias the representation. Do not compute statistical descritors with the censoring criterion indicated as it artificially bias data towards lower values.
Figures 5 and 6 are meaningless. Authors should explore alternative ways to represent their findings (add concentrations of like-compounds to show trends, select representative compounds, etc. On the other hand, what is the source of PCB153? The simple observation of the presence of a compound is not as important as to try to reveal why this compound is present in the system (and its source)
The figures have been removed and replaced by a simpler representation of the PAHs and PCBs sums. A table with these sums has also been added to the Supplementary Information. The potential sources of PCB-153 have been discussed in the article.
In page 17, line 364. Marine aerosol data is not presented. It is advised to perform a study about the contribution of precipitation to the local surface water studied.
The former research in the area (especially of ionic compounds in precipitation) indicates strongly, that marine aerosol is an important chemical factor in Hornsund.
The PCA analysis presented does not include information about the scores obtained. The reported weights in the variance of the different components are nothing what cautions against its representativity and significance.
The PCA scores are supplied now in a supplementary csv file.
Reviewer 2 Report
Abstract-
No need to describe all parameters presented in this assay (all metals, all organics etc.), it should merely describe the targets, tasks, results and conclusions (in a few lines).
Introduction-
Lines 36- 46 are a part of the discussion
The introduction is very short, no description of a worldwide literature or any impact on the area is described.
The hyperlink in the paper does not work
Materials and methods-
I suggest moving great parts of the sites history and descriptions- to the introduction.
The mentioning of the previously published data seems to be of great importance.
Lines 124-127:
“Samples were of various size and thus only allowed for analysis of selected compounds, since they were originally collected for various studies and only afterwards the collected information was pooled for the wider spatial comparison presented here. Thus the number of determined samples and specifically addressed sampling objects is not the same for various compounds.”.
Please explain- what exactly do you mean … are different samples collected or extracted in different ways?
Lines 164-165:
A SIM mode seeks for more than one ion mass per compound (at least 3), these ions should also have a very strict ratio in order to confirm the presence of a specific compound. Thus I wonder if the analysis was carried using only one ion per compound in a SIM mode?
Many parts and sections such as the CRM approval, ISO certification etc.. in addition to table 1 which is merely a validation for the methods used, should in my opinion be included in the supplementary section only. They are not the main issue of this paper and are making the reading experience a tedious job.
Results:
Figures are way too complicated, I strongly recommend using a table .
Figure 5- not readable at all. The index is missing, it seems like a total mess (I suggest a table, at least in the supplementary section). Including the outliers, might be possible via using a special y-axis that has a heterogeneous distribution of numbers (instead of linear), an option frequently used to include data points much greater than most points.
Figure 6: just for clarification – since most of the boxes are below 0.05ng/L, are they all below the detection limit for PCBs (0.075 ng/L)?
Discussion:
Line 369: do you believe that the pH and conductivity record of the sea spray is apparent in the water mass (were sampled- at 0.5 m in high volume water bodies)?
Line 384: the conclusion seems ok for PC1 and PC2 but not that clear for PC3 and PC4 chart. When you rely on PCA- on which parameters do you mean?
Line 441: do you mean PCBs, PAHs…numbers below LOQ?
If possible – correlate them to the distance from the station. Outliers are also very important.
You can also correlate them to air mass streams at the specific dates of sampling and in different altitudes (see for example - https://doi.org/10.1016/j.marchem.2018.07.009)
In my opinion, regarding the conclusions about the outliers and PCBs+PAHs remains, these last remarks are crucial…
Finally, since as stated the data was all collected at the same season, I wonder if we can project the results here to the entire year? please explain in details what are the conclusions from this project/s? - the conclusions section is very limited.

Author Response
Dear Reviewer,
many thanks for the careful reading and consideration given to our article. In light with your remarks, we have prepared a new version of the manuscript for your perusal, focusing especially on the rigour and readability of our work.
With respect to major issues highlighted by both Reviewers, we have applied the following changes:
The Introduction has been changed to include parts of the more generalised field site description, as well as the literature presenting the wider than local importance of such work undertaken in the vicinity of the Polish Polar Station Hornsund. We maintained the reference to „local sources” for elevated concentration of certain elements and compounds from natural sources (however, they are not named „pollution” directly in text). Although it would have been more accurate to calculate the geochemical background, we have limited information available which would be necessary for this procedure to be conducted rigorously. In Materials and Methods, more information is provided about the novel elements in this work as opposed to the once-published and now pooled data. The details of analytical quality assurance and quality control have been moved to a Supplementary Information In response to concerns as to the interpretation of the principal component analysis (PCA), in the context of the use of values below the detection limit with a censoring procedure of replacing those with ½ LOD, we have performed a sensitivity analysis. It is presented in the Supplementary Information We have also supplied PC scores for the main analysis occurring the article (as a supplementary file in csv format). The discussion of the sources of pollution has been expanded, including information about the potential input from precipitation and snowmobile exhaust. We have also removed the reference to WHO limits as not relevant to the study subject.At the correction stage, we have also noticed a mistake (ommission of a few data points from Fuglebekken lake for SCFAs, PAHs and PCBs).We sincerely apologise for any inconvenience caused by this. Now the whole article has been corrected to include this part of the dataset.
Responses to specific comments:
Abstract
No need to describe all parameters presented in this assay (all metals, all organics etc.), it should merely describe the targets, tasks, results and conclusions (in a few lines).
The abstract has been rewritten.
Introduction
Lines 36- 46 are a part of the discussion
The indicated extract has been moved to the Discussion.
The hyperlink in the paper does not work
The hyperlink has been removed.
Lines 124-127:
“Samples were of various size and thus only allowed for analysis of selected compounds, since they were originally collected for various studies and only afterwards the collected information was pooled for the wider spatial comparison presented here. Thus the number of determined samples and specifically addressed sampling objects is not the same for various compounds.”.
Please explain- what exactly do you mean … are different samples collected or extracted in different ways?
Lines 164-165:
A SIM mode seeks for more than one ion mass per compound (at least 3), these ions should also have a very strict ratio in order to confirm the presence of a specific compound. Thus I wonder if the analysis was carried using only one ion per compound in a SIM mode?
In the initially submitted manuscript, we gave ions based on which concentrations were calculated. Now, in Table S2, we list the ions used to determine the concentration [Ion 1] as well as two confirming the presence of the compound.
Results:
Figures are way too complicated, I strongly recommend using a table.
Figure 5- not readable at all. The index is missing, it seems like a total mess (I suggest a table, at least in the supplementary section). Including the outliers, might be possible via using a special y-axis that has a heterogeneous distribution of numbers (instead of linear), an option frequently used to include data points much greater than most points.
Figure 6: just for clarification – since most of the boxes are below 0.05ng/L, are they all below the detection limit for PCBs (0.075 ng/L)?
The Figures 5 and 6 have been removed and replaced by a simpler representation of the PAHs and PCBs sums. A table with these sums has also been added to the Supplementary Information. However, we have retained the other figures – a data file, equivalent to a table, is supplied as a supplement to the article.
Discussion:
Line 369: do you believe that the pH and conductivity record of the sea spray is apparent in the water mass (were sampled- at 0.5 m in high volume water bodies)?
Please note the shallow water bodies have been sampled at a smaller depth, that many of the sampled lakes were shallow and located very close to the sea. However, sea salt may also be an influence of melting permafrost from the uplifted sea terrace upon which these lakes are located.
Line 384: the conclusion seems ok for PC1 and PC2 but not that clear for PC3 and PC4 chart. When you rely on PCA- on which parameters do you mean?
When interpreting the PCA, we started with the correlation of PCs with variables and included the visible clustering of study objects in the presented graphs. Please note that the graphs give insight into both the PCA loadings (through the variable arrow positions) and scores (through data point coordinates), thus a visual interpretation may be very informative. However, we have limited our interpretation of the PC3-4 space because it seems more sensitive to the chosen treatment of data <LOD. The presented graphs may now be compared with a sensitivity analysis presented in the Supplementary Information.
Line 441: do you mean PCBs, PAHs…numbers below LOQ?
The sentence has been rewritten to: „Due to the smaller subset of samples providing the concentrations of PAHs, PCBs and SCFAs, these chemicals have been excluded from the PCA.” We did not mean the numbers below LOQ, but the whole pool of analysed samples, whether they yielded results below or above LOQ.
If possible – correlate them to the distance from the station. Outliers are also very important.
You can also correlate them to air mass streams at the specific dates of sampling and in different altitudes (see for example - https://doi.org/10.1016/j.marchem.2018.07.009)
None of the so far presented samples come from precipitation, thus they cannot be directly connected to air circulation patterns. If the impact of the locally pervasive sea-spray causes doubts, then the interpretation of stream and lake waters in the context of air mass trajectories is even more risky, since one precipitation event from a particularly contaminated air mass may have a stronger signal than an average of all precipitations across the rest of the month. Furthermore, across the small represented area the air mass trajectories should not differ significantly, thus they would not offer an extra explanation for the observed spatial patterns. What may differ is the deposition from such air masses, e.g. through the cold condensation effect, which is related to topography, and this may be observed in the case of PCB-153 and PCB-180. But again, the waters we study here present a more averaged chemistry to single precipitation events.
In my opinion, regarding the conclusions about the outliers and PCBs+PAHs remains, these last remarks are crucial…
We have expanded the analysis of these compounds to the extent which we deem robust, taking into account the properties of the dataset. We hope that the corrected paper will satisfy this comment, too.
Finally, since as stated the data was all collected at the same season, I wonder if we can project the results here to the entire year? please explain in details what are the conclusions from this project/s? - the conclusions section is very limited.
The seasonal change aspect has been added to the concluding paragraph, as were other considerations.
Round 2
Reviewer 1 Report
The reviewer kindly acknowledges the significant effort made by the authors to try to follow (when possible) the suggestions provided in the preliminary assessment. The answer given to the questions, text reordering (and shift of some part to supplementary materials) as well as the clarifications introduced improve the overall quality of the manuscript and makes it sounder. The reviewer considers that the modifications are satisfactory. As a suggestion, the reviewer would draw the attention to try to improve the readability of PCA plot (Figure 8) due to color selection and label overlapping. One way could be to separate symbols from the vector lines, presenting the last ones as a panel inset within the plots symbol plots.
In any case, the reviewer congratulates the authors for the good work performed.
Author Response
Many thanks for the consideration given to our paper. We will contact the editorial team about the potential corrections for figure clarity.
Reviewer 2 Report
Regarding the text-
The abstract is well written.
The introduction- short, but fine.
Several remarks regarding the rest:
The description of the study site should be moved to the introduction... I'ts too long. Materials and methods study site section- shouldn’t be a written book but a short description of the site.
Line 143- different water volumes does not matter due to the fact that chemistry figures are always reported as concentration (per volume of water or kg sediment) and thus I don’t understand.
Line 212: "For statistical analysis (i.e. in the dataset provided in Supplementary Information), all 212 values below the limit of detection (LOD) were replaced with half of its value"- depends on how many points these are- if the great majority are below the LOD, using their values may cause a bias (don’t have the supplementary data).
Lines 477-480- sea spry or permafrost melting - elevating the EC, I saw the reference regarding snow melting and Na, Cl and sulfate residues. can you evaluate the contribution of these to the total volume of shallow lakes water? mainly of the aerosol?
Regarding the authors answer: " None of the so far presented samples come from precipitation, thus they cannot be directly connected to air circulation patterns. If the impact of the locally pervasive sea-spray causes doubts, then the interpretation of stream and lake waters in the context of air mass trajectories is even more risky, since one precipitation event from a particularly contaminated air mass may have a stronger signal than an average of all precipitations across the rest of the month. Furthermore, across the small represented area the air mass trajectories should not differ significantly, thus they would not offer an extra explanation for the observed spatial patterns. What may differ is the deposition from such air masses, e.g. through the cold condensation effect, which is related to topography, and this may be observed in the case of PCB-153 and PCB-180. But again, the waters we study here present a more averaged chemistry to single precipitation events"
-
The precipitation of one or more events of aerosol (dust+ seaspry +....) may not be the only component but a rather origin that may explain some of the results. Yes the cold condensation theory should support the higher concentrations of volatile PCBs yet it does not take into account the major uses of some pcbs above others. I guess if the major component was airborne particles, I agree that you should have seen low differences yet you can not rule that out, I guess a good advice would be combining a very simple setup of dust/sea spray collectors.
End .
The bottom line is- though I don’t have the supplementary section, my impression is that a lot of data processing is suffering from flaws- e.g. points below LOD, snow vehicles tracks and I guess It is not ready for publication yet. I would shorten and eliminate the fancy statistics and charts that in my opinion are not readable and somewhat tedious...
Author Response
Many thanks for the thorough consideration given once more to our work. We address the listed concerns below in a point-by-point manner, hoping we will clarify the doubts. Unfortunately, we were not able to apply all the suggested changes, and we provide the details for our decisions below.
Regarding the text-
The abstract is well written.
The introduction- short, but fine.
Thank you - no changes were made.
Several remarks regarding the rest:
The description of the study site should be moved to the introduction... It's too long. Materials and methods, study site section - shouldn’t be a written book but a short description of the site.
We do not feel that such technical information fits in the Introduction (because it’s too specific). On the other hand, removing the field site information would contradict the recommendation from the Water journal to disclose „full methodical and/or experimental details” and „as much detail as possible”, as is stated in the Aims & Scope of the journal. In the context of the special issue focus on surface catchments and the local and global distribution of pollutants, we feel obliged to provide a comprehensive site description as an essential part of our work.
Line 143- different water volumes does not matter due to the fact that chemistry figures are always reported as concentration (per volume of water or kg sediment) and thus I don’t understand.
The water volume matters due to technical constraints of the analytical procedure (a division of the sample into aliquots for multiple procedures). The sample analyses we conducted are destructive, thus the same water volume cannot be reused to provide analyses of two different classes of chemicals, with different preparation procedures. Furthermore, a small sample volume in the case of PAHs and PCBs analysis significantly increases the detection limits, rendering the analysis not very informative if performed in a small sample volume.
Line 212: "For statistical analysis (i.e. in the dataset provided in Supplementary Information), all values below the limit of detection (LOD) were replaced with half of its value"- depends on how many points these are - if the great majority are below the LOD, using their values may cause a bias (don’t have the supplementary data).
Whenever there was a problem with a high number of results not exceeding the LOD, we mention this in the manuscript (in lines 260, 339, 346, 352, 354 and 367 - numbering from the manuscript version after the first revision round). This is the best we can do - except asking the Editor to provide you with the full data table which is a supplement to this work, and is intended as part of the publication.
Lines 477-480 - sea spray or permafrost melting - elevating the EC, I saw the reference regarding snow melting and Na, Cl and sulfate residues. Can you evaluate the contribution of these to the total volume of shallow lakes water? Mainly of the aerosol?
Thank you for this comment, indeed the data is scarce but we will attempt to assess whether such a contribution may be significant.
There is only incidental data on the total ionic deposition in precipitation in Svalbard, and the data from Hornsund are, as yet, unpublished (although we know they are part of the monitoring programme of the station). In southern Svalbard (Calypsobyen and Hornsund), Krawczyk et al. [1] estimated summer rains to contribute between 121 and 345 mg m-2 of acidifying ions (SO42− + NO3−), while base cations (Ca2+ + Mg2+) in Calypsobyen were deposited at a load of approximately 263 mg m-2 and the main components of sea salt (Cl- + Na+) reached 753 mg m-2 deposition. The total annual loads of inorganic ions in snow on the nearby Hans glacier may exceed 8000 mg m-2 (C2S3 project consortium, article in preparation).
The small, shallow tundra lakes – many of the water bodies studied by us – are typically between 30 and 100 cm maximum depth, and we would approximate their area-averaged depth at 50 cm due to the relatively small extent of their deepest parts. Thus the deposition of 1000 mg m-2 of sea salt would disperse in approximately 500 L of lake water, increasing their concentration by 2 mg L-1. This is a negligible signal compared to some surface water ionic concentrations exceeding 100 mg L-1 and a small but detectable signal in the more dilute lakes described by Ntougias et al. [2], which exhibited TDS concentrations of 30 mg L-1. We appreciate that only in a non-typical situation could the atmospheric signal be detected as variability in the lake water concentrations in this area. Thanks to your comment, we consider now the saline groundwater hypothesis a more likely explanation for our observations regarding EC, and this fragment has been edited accordingly.
Regarding the authors answer: "None of the so far presented samples come from precipitation, thus they cannot be directly connected to air circulation patterns. If the impact of the locally pervasive sea-spray causes doubts, then the interpretation of stream and lake waters in the context of air mass trajectories is even more risky, since one precipitation event from a particularly contaminated air mass may have a stronger signal than an average of all precipitations across the rest of the month. Furthermore, across the small represented area the air mass trajectories should not differ significantly, thus they would not offer an extra explanation for the observed spatial patterns. What may differ is the deposition from such air masses, e.g. through the cold condensation effect, which is related to topography, and this may be observed in the case of PCB-153 and PCB-180. But again, the waters we study here present a more averaged chemistry to single precipitation events"
The precipitation of one or more events of aerosol (dust + seaspray +....) may not be the only component but a rather origin that may explain some of the results. Yes the cold condensation theory should support the higher concentrations of volatile PCBs yet it does not take into account the major uses of some PCBs above others. I guess if the major component was airborne particles, I agree that you should have seen low differences yet you can not rule that out, I guess a good advice would be combining a very simple setup of dust/sea spray collectors.
There is only a little bit more data we can offer to address this criticism, which we have not included in the last review round due to time constraints. We have fragmentary data on the precipitation composition in the area, collected during the same field season (in summer 2010). It is only a few samples, yet they may shed light on the origins of the pollutants found in surface waters. Please find these considerations in the revised manuscript’s Supplementary information, in the subsection S3. We have also added the lines 466-474 in the corrected version of the manuscript, relating to that.
We agree that a valuable future research idea in this location is to employ passive or active samplers to measure the atmospheric concentrations of various pollutants. In fact, some of the authors of this study are currently conducting a project which includes such a sampling of PCBs.
The bottom line is - though I don’t have the supplementary section, my impression is that a lot of data processing is suffering from flaws - e.g. points below LOD, snow vehicles tracks and I guess it is not ready for publication yet. I would shorten and eliminate the fancy statistics and charts that in my opinion are not readable and somewhat tedious...
We think that the main misunderstanding here may be caused by the lack of availability of the supplement to the Reviewer. Hence, we will request with the Editor that all three supplementary files are provided together with the article for the next round of review. We have provided some data addressing the first round of reviews in there and we hope that given the full scope of our work, the Reviewer's assessment would differ.
We feel that we have thoroughly addressed the concerns about values below LOD and also that this is a common problem with environmental chemical data – otherwise many datasets would not be usable at all.
Regarding the snow vehicle tracks, the topic is actually a frequently omitted element in surface water data interpretation, while we have at least incidental data on those, which are worth publishing as a rare piece of information.
Finally, please consider that statistics is our main tool in this work, thus it would be difficult to eliminate it from the manuscript. Besides the one multivariate method (PCA), the methods we used are very basic and intended to express numerically whether what we see as a pattern is significant and worth further consideration or not, thus providing a more objective stance than the visual assessment of graphs. In our opinion, the statistics used are safeguarding against misleading human intuition in the interpretation of the data. Thus removing them would affect the rigour of our work.
Krawczyk, W.E.; Bartoszewski, S.A.; Siwek, K. Rain water chemistry at Calypsobyen, Svalbard. Polish Polar Res. 2008, 29, 149–162. Ntougias, S.; Polkowska, Ĺ».; Nikolaki, S.; Dionyssopoulou, E.; Stathopoulou, PDoudoumis, V.; Ruman, M.; Kozak, K.; NamieĹ›nik, J.; Tsiamis, G. Bacterial Community Structures in Freshwater Polar Environments of Svalbard. Microbes Environ. 2016, 31, 401–409.
Round 3
Reviewer 2 Report
I dont have any remarks apart from a few general recomendations: I would reconsider some of the last mentioned comments sent. Specifically, though I know that as you answered, many papers use an estimation of the "results below the LOD" only to conduct a statistical survey, I think that at the end of the day- results lower than the detection level are no results at all (and its better to depend on other results). On the other hand I guess that if other papers use this trend of analysis, its up to the editors decision .
Author Response
We appreciate the Reviewer's concern for the interpretation of results below the limit of detection. However, we strove to be careful in the data interpretation and we clearly state to the reader what has been done with the data to obtain the conclusions found in the paper. In this form, we claim it better to report such results than to let them remain unpublished. If openly available, our data and analysis can guide future attempts to constrain the impurity concentrations in Svalbard waters and their sources, which would be very welcome.